# Changes in Body Composition and Nutritional Periodization during the Training Macrocycle in Football—A Narrative Review

**DOI:** 10.3390/nu16091332

**Published:** 2024-04-29

**Authors:** Wiktoria Staśkiewicz-Bartecka, Marek Kardas, Grzegorz Zydek, Adam Zając, Jakub Chycki

**Affiliations:** 1Department of Food Technology and Quality Evaluation, Department of Dietetics, Faculty of Public Health in Bytom, Medical University of Silesia in Katowice, ul. Jordana 19, 41-808 Zabrze, Poland; mkardas@sum.edu.pl; 2Department of Sport Nutrition, Jerzy Kukuczka Academy of Physical Education in Katowice, ul. Mikołowska 72A, 40-065 Katowice, Poland; g.zydek@awf.katowice.pl; 3Department of Sports Training, Institute of Sport Sciences, Jerzy Kukuczka Academy of Physical Education in Katowice, ul. Mikołowska 72A, 40-065 Katowice, Poland; a.zajac@awf.katowice.pl (A.Z.); j.chycki@awf.katowice.pl (J.C.)

**Keywords:** football nutrition, body composition, macrocycle training, periodization of nutrition

## Abstract

Nutrition periodization in football training is an important determinant of adaptation to cyclic training loads. Personalizing an athlete’s diet is crucial to ensure optimal performance and body composition, depending on the phase of training. The purpose of this review is to answer the question of how the body composition of football players changes over the training macrocycle and how dietary recommendations should be tailored to specific training periods. The review of scientific evidence was conducted based on the available literature, typing in phrases related to training and nutrition periodization using the PubMed and Google Scholar database methodology tools. A literature search resulted in the selection of 346 sources directly related to the topic of the study, and then those with the highest scientific value were selected. There is a need to adjust energy and nutrient intake according to the different training phases in a football player’s preparation cycle. During the preparatory phase, it is recommended to increase protein and energy intake to support anabolic processes and muscle mass development. During the competitive period, due to the intensity of matches and training, the importance of carbohydrates for glycogen replenishment and recovery is emphasized. The transition phase requires the regulation of caloric intake to prevent adverse changes in body composition. Hydration has been identified as a key element in each phase of training. Cooperation between coaches, nutritionists, and players is essential to optimize sports performance and rapid recovery, and the authors recommend continuous adaptation and nutritional optimization as an integral part of football training.

## 1. Introduction

One of the most popular sports right now is football, which is drawing record crowds to stadiums and TV screens. According to the FIFA Big Count 2006, about 270 million individuals, or 4% of the world’s population, play football [1]. The FIFA 2022 World Cup in Qatar involved over 800 players from 32 countries, with a total estimated market value exceeding EUR 12 billion [2]. Despite the sport’s popularity and the substantial amounts of money invested in it, players still have insufficient nutritional awareness [3]. Athletes have been allowed to eat whatever they wanted, whenever they deemed it suitable, for a long time. However, the increasing demands on athletic ability, the rise in the pace and intensity of football games over the past 20 years, and the commercialization of the sport have led to a search for solutions to enhance the players’ athletic level. One approach to achieve this is to adjust nutrition to meet the demands of football [4,5].

In the last few years, concerns about athletes’ diets have received a lot of attention. A well-balanced diet that considers energy, macronutrients, fluids, and the right supplements can positively impact sports performance [6]. The nutritional aspect constitutes an important element in an athlete’s training program. International guidelines, grounded in scientific research, advocate for the appropriate amount, type, and timing of food intake to ensure effective training, while minimizing the risk of injury and trauma [7]. Achieving optimal metabolic efficiency requires striking a balance between nutrition, training, and the recovery process. Providing energy from suitable sources and maintaining a proper energy balance are crucial for individuals leading active lifestyles, particularly professional athletes [8].

The purpose of this study is to examine the concept of nutritional periodization within the football training macrocycle and its efficacy in managing diet to optimize the athletic performance and body composition of football players. This includes analyzing the adjustments needed in energy intake, macronutrient ratios, and hydration across different phases of the training cycle to individual players’ athletic and health objectives.

## 2. Method

The main aspect that motivated the review work carried out was to look for changes in body composition occurring during the various elements of the training macrocycle in the works cited, and to find nutritional recommendations to support these changes. Unfortunately, the current state of knowledge on the subject, despite the many studies, is still poor, so the authors decided to conduct a review of the most current knowledge in this area, to identify those sources that address the described topic and gather in one place the available knowledge.

### 2.1. Review Procedure

A preliminary search for items consistent with the topic and purpose of the review was conducted to identify the research field. After reviewing the existing data, a keyword package was selected that seemed most relevant and consistent with the review topic.

The databases PubMed and Google Scholar were searched using the following (MeSH)terms: “body composition analysis”, “macrocycle”, “preparatory period”, “competitive period”, “transition period”, “football”, “soccer”, “nutritional recommendation”, “nutrition in football”, and “supplements”. The MeSH terms were internally validated by the coauthors. A review of the scientific evidence was conducted based on the available literature by entering sample phrases (consistent with the MeSh dictionary) with Boole operators, logical operators, and special characters.

### 2.2. Eligibility Criteria and Search Strategy

Articles in this review had to be (1) published before the end of December 2023 and after December 2003, (2) available as a full text in English, (3) categorized as original research, reviews, or meta-analyses, and (4) the studied population comprised adult football players (aged 18 years). Articles that were published after 2003 have been included to make sure that the topic covered is not a completely new field of research, but also to avoid very old data, because dietetics, as well as the physical demands of football, are rapidly evolving. The database review was completed on 1 March 2024. Titles and abstracts were reviewed to verify these criteria. If all inclusion requirements were met or remained unclear, articles were read in full. If the full text showed that not all the requirements were met, the article was excluded. Additional literature was obtained by searching references in manuscripts (snowball method).

A literature search yielded 346 sources directly related to the topic of the study, and then those with the highest scientific value were selected according to the eligibility criteria.

### 2.3. Critical Appraisal

When assessing the sources critically, consideration was given to whether the publications had been published in peer-reviewed journals with at least two reviewers. As previously mentioned, 138 sources were eligible for the final review. The primary restriction of the employed methodology was the omission of sources composed in languages other than English.

“Gray literature,” or works that have not undergone review or are university-owned (theses, conference reports, government pamphlets, newsletters, etc.), was not included in the review. These sources are valuable, but there is a good chance they include out-of-date information.

## 3. Characteristics of the Sport of Football

Sports usually involve at least two opponents, often organized into teams, pairs, or individuals that work together to achieve common aims. Since the primary purpose of football is to score a goal and keep the opposition from scoring a goal, this leads to the emergence of tactical behavior and strategic definitions [9,10]. Since the aims of the two teams are mutually exclusive, there are dynamic interactions between the parties, which evolve during the game, leading to a process of interaction [10,11]. 

Football is a high-intensity activity, where periods of active recovery or passive rest are used to offset the exertion. The effort during a match is variable, involving running, sprints, and dynamic changes of direction, during which players perform about 726 different movements and turns [12,13]. At the highest level, field players travel roughly 10–12 km, while goalkeepers go about 4 km [14]. Studies report that players in the midfield position cover the longest distances during the game [15,16]. The intensity of effort decreases in the second half of the match, as does the distance, which is reduced by 5–10% [17]. During a football game, sprinting occurs every 90 s or so and lasts an average of 2–4 s [14]. Football players walk for about 18–27 min (20–30% of game time), run at very low intensity for about 13–23 min (15–25% of game time), run at moderate intensity for 9–13 min (10–15% of game time), and run at high intensity for 4–7 min (4–8% of game time) [15,17]. From the point of view of the intermittent effort profile, the work-to-rest ratio is 1:8 [17]. Each player engages in 1000–1400 short activities throughout a match, changing every 4–6 s [14].

## 4. The Importance of Body Composition in Football

The body composition of athletes is determined by the proper supply of nutrients, which results in adequate performance of the body [18,19]. An athlete’s body composition can provide valuable insight into their general health and physical performance [20]. Age, gender, somatotype, genetics, degree of physical activity, and individual variability all influence the ideal body composition [21]. Body composition in athletes affects movement quality and performance levels, which in turn affects athletic performance, in addition to being viewed as a health condition [22]. 

Anthropometric and fitness traits that have been linked to specific football players’ success have been found in recent years [22]. It has been confirmed that there are predispositions directed toward players performing different functions on the field during games [23]. In football, four playing positions are distinguished: goalkeeper, defender, midfielder, and striker. Accordingly, players are characterized by specific anthropometric and performance characteristics that allow them to achieve better results. Goalkeepers and defenders are characterized by a higher height and greater body mass than other players [24]. Compared to strikers and midfielders, defenders typically have larger bodies and are taller. The smallest group of players in terms of body mass and height is the midfield [25]. When comparing the body composition of football players, it is crucial to take relative values (e.g., %FM instead of FM [kg]) into account, since variations in body size may be the cause of the variance in absolute values of lean body mass between players in different roles, rather than variations in body composition [23,25]. Due to the varying physiological and metabolic demands of football depending on the function played on the field, players’ body compositions varies [26].

The optimal physique of a player, taking into account body fat mass (FM), lean body mass (LBM), and fat-free mass (FFM), depends on his function on the field and individual predispositions [22]. There is no single value that is a reference for percent body fat (%FM), but the average level of it in the body of football players takes values in the range of 8% to 13%, although values deviating from the above have been described [27,28].

Football players may need to make adjustments to the daily ratio’s composition at different periods during the season to maintain an ideal body composition [18]. This could happen in the preseason or after an injury. Strength and power can be enhanced by increasing skeletal muscle mass, which is a desirable training adaptation [29]. In addition, preserving skeletal muscle mass during injury and immobilization projects a faster return to full function. Excess body fat negatively affects an athlete’s mass-to-power ratio, acceleration ability, and total energy expenditure [30]. Athletes may also choose to consciously alter their body composition to achieve a desired appearance, which can interfere with athletic goals [11]. Therefore, each football player’s body composition goals should be agreed upon with a nutritionist or team physician [11,27].

## 5. Time Structure of the Training Process and Physical Requirements and Changes in Body Mass Composition

An athlete should view training as a process that prepares them physically, mentally, strategically, and technically to perform at their best [31]. This multifactorial approach maximizes the effects of the training stimulus by utilizing established physiology, psychology, and physics concepts [32]. Since the goal of training is to increase performance, it must include a sufficient stimulus for adaptation, appropriate methods for tracking improvements, and other pertinent measures like rest, mental support, healthy eating, and carefully chosen supplements [33].

Maintaining an optimal body mass is essential for overall health [27], particularly for individuals involved in increased physical activity, like professional athletes. The body composition of players is influenced by the physical stress caused by professional football [13,14]. Research indicates that factors like nutritional knowledge, food preferences, and level of activity help athletes maintain their ideal body composition [7,34,35,36,37,38,39,40]. Athletes’ current training programs are primarily structured according to the periodization theory [41], which involves the deliberate sequencing of various training units to meet predetermined goals [42,43].

The amount of work an athlete completes within the examined time frame, training unit, or training cycle refers to their training load [44]. The concept of load comprises two values: volume and intensity of effort. Volume represents the quantitative component of work, expressed in terms of time, distance, weight, and number of repetitions. Intensity, on the other hand, is the component of work resulting from the speed of the exercise performed, the number of repetitions in a given unit of time, the number of series, and the nature of intervals. Therefore, it is expressed by the ratio of power developed to the maximum power achievable in a particular exercise by a given athlete [43,44]. The variation in physical loads between athletes is extremely important and is related to the specificity of the various functions performed on the field during football matches [15,16,17,23,37,45]. A direct response to the demands of the function performed is the energy cost to the body, which modulates the composition of the body during the season [46,47].

An essential tool for getting athletes ready for competition is body composition analysis, which also shows a player’s nutritional status. Optimal body composition is an important component of fitness, as excess body fat acts as unnecessary weight in the typical activities of the game, i.e., running and jumping. It is known that body fat content influences acceleration capacity, power-to-weight ratio, and energy expenditure [23]. Conversely, each functional performance parameter’s value is determined by the percentage of lean body mass [48].

The time structure of football training takes into account various aspects, such as the development of physical endurance and the optimization of training processes [48]. Depending on the competition calendar, the preparation of players in team sports is carried out in single-cycle, two-cycle, and three-cycle models. In professional football, a two-cycle training model is usually used. The two-cycle model of the training process involves competitions in the spring and fall system. One cycle is a six-month macrocycle that is known as the fall round, which runs from July to December, and the other is the spring round, which runs from January to June [48]. In football, three training periods function in the structure of the six-month macrocycle: preparatory, competitive, and transition. Macrocycle training in football involves planning the entire training process of professional players over a longer period. The fundamental cycle in professional football clubs’ training is the macrocycle with the aforementioned periods [41]. Figure 1 shows a diagram of the training macrocycle in football.

### 5.1. Preparatory Period

Fundamental to the construction and stabilization of the team’s sporting form are the preparation periods, occurring in the spring (January and February) and autumn (second half of June and July) [49]. Unlike individual sports, football’s preparatory period is relatively shorter compared to the competitive period, a disproportion that especially intensifies when teams participate in international competitions [50]. During this phase, the goal of training is to restore athletes’ physical condition following the transition period, with training loads generally increased compared to the competitive period [51,52]. Intensive conditioning training, friendly matches, and technical and tactical exercises are scheduled in parallel during this period, aiming to enhance the body’s ability to exercise, as well as its technical, motor, and tactical skills [52]. Examples of exercises include passing drills, dribbling courses, shooting practices, defensive shaping possession games, set-piece training, and position-specific exercises [11].

The season’s period significantly influences an athlete’s body composition. During the preparatory period, the athletes’ fitness is shaped and performance recovers from a period of rest [53]. Modifications to training loads and nutrition are made to align with the athlete’s desired outcome during this phase, aiming for an optimal loss of fat mass (FM), while preserving or increasing lean body mass (LBM), to impact sports performance positively [54,55].

A study by McEwan et al. showed a reduction in FM during this period from 10.6 ± 1.88 kg to 9.56 ± 1.81 kg and an increase in LBM from 59.58 ± 5.27 kg to 60.61 ± 5.18 kg in elite Spanish league players [28]. In a study by Owen et al. conducted with professional European football players during the preparatory period, an increase in LBM from 67.21 ± 5.31 kg to 68.23 ± 5.49 kg and a reduction in FM from 7.47 ± 1.78 kg to 6.79 ± 1.54 kg were observed [37]. The results obtained in the study by Staśkiewicz et al. indicate an increase in SMM content from 40.95 ± 3.48 kg to 41.3 ± 3.53 kg on average during the preparatory period [56]. Analyzing the results of Staśkiewicz et al. on the preparatory period, taking into account the function performed by the player during the match, they showed a significant increase in LBM in defenders from 73.94 ± 4.67 kg to 74.93 ± 4.16 kg, and an increase in SMM from 42.63 ± 2.79 kg to 43.27 ± 2.53 kg. In addition, SMM content increased from 38.96 ± 3.42 kg to 39.2 ± 3.39 kg in midfield players [56]. A study by McEwan et al. confirms these results; during the preparatory period, defenders increased their LBM from 61.83 ± 4.58 kg to 62.49 ± 4.47 kg, while that of midfielders increased from 57.36 ± 2.72 kg to 59.81 ± 3.1 kg [28]. A study by Owen et al. found an increase in LBM and a reduction in FM during the preparatory period in players serving as defenders, midfielders, and forwards [37].

### 5.2. Competitive Period

Increasing aerobic capacity is the primary goal of training during the competitive period, leading to more running activity and less movement time during games [57]. This phase presents challenges in planning, as training loads must be adjusted to maximize physiological adaptations, while avoiding overtraining and injury [49,58]. The competitive period focuses on maximizing athletic performance based on the skill dispositions and fitness levels achieved during the preparation period [6]. Elite players experience a heavy load during this period due to participating in more matches [13,59].

Training loads during the competitive period encompass both external and internal factors. External loads relate to physical exertion during training or matches, while internal loads involve psychological factors and biochemical stress responses [46]. Despite advancements in understanding the physical demands of football, the detailed tracking of players’ training loads remains an area of active research [12,60]. Studies indicate differences between the intensity of training and match loads, with training sessions covering shorter distances at lower intensities compared to matches [12,46,61].

In a conventional seasonal scheme with one game per week, players often conduct 4–5 training sessions on the field, distributing the total training load according to the day of the week and the priority of the upcoming match. Additionally, players engage in off-field training sessions, such as weight training, resulting in a total of 8 to 10 training sessions per week [44]. Resistance training may be tapered during this period to maintain strength without causing fatigue, utilizing about 50–70% of 1 RM, with an emphasis on explosive movements to promote power. The Rate of Perceived Exertion (RPE) for overall training sessions might range from moderate to high (6–8 out of 10), with individual drills potentially reaching higher RPEs of 9–10 but with sufficient recovery to ensure readiness for match play. Coaches aim to peak players’ physical and technical abilities during this period, while ensuring they are fresh for competition [62].

However, in the competitive period, training intensity is reduced compared to the preparatory period, with more focus on recovery training between games [39]. Professional athletes adjust their diets to match the exercise volume during this time, aiming to maintain a suitable body composition throughout the competition. Imbalances in energy consumption can lead to fluctuations in body fat and muscle mass, impacting both performance and health [63,64]. It is important to note that the organization of training sessions, both strength and on-field, may not follow a strict sequence, potentially affecting athletes’ varying macronutrient intake profiles [59].

Analyzing body composition during the competitive period, a study by Staśkiewicz et al. found a decrease in LBM from 71.68 ± 5.85 kg to 71.14 ± 5.93 kg and an increase in FM from 9.76 ± 2.57% to 10.32 ± 2.77% [56]. A study by Carling et al., involving 30 professional football players participating in the French League 1, showed an increase in LBM over the competitive period from 69.41 ± 5.53 kg to 70.1 ± 5.67 kg. FM content, on the other hand, was reduced from 10.45 ± 1.61% to 10.19 ± 1.75% [23]. A study by Kultu et al. found a reduction in LBM content from 63.2 ± 6.0 kg to 62.6 ± 6.0 kg and a reduction in FM content from 10.2 ± 3.0% to 10.0 ± 3.0% [65]. Devlin et al. conducted a study involving 18 Australian football players and showed a reduction in LBM from 56.8 ± 5.1 kg to 56.4 ± 5.5 kg on average during the competitive period, and an increase in FM from 8.7 ± 1.4 kg to 9.5 ± 1.7 kg on average [54]. One possible explanation for the concurrent decrease in LBM content and rise in FM content during the competitive period could be a reduction in training load. General training and high-intensity conditioning exercises are part of the training program during the preparatory phase, whereas game tactics, ball possession, and fixed game passages comprise a significant amount of the competitive period’s workouts and are associated with lower workloads. These findings imply that the players’ body composition cannot be maintained by match effort alone [66].

### 5.3. Transition Period

During the transition period, training participation is either drastically reduced or completely stopped, allowing athletes to engage in voluntary, non-periodic training or recreational sports [64]. This period is primarily focused on treating injuries of various severities and readapting the body to apply similar training stimuli in the next competition cycle [13]. However, this cessation of training can result in the partial or total loss of some evolved adaptive alterations brought about by the preparation and competition periods [67]. Detraining, which is the partial or complete physiological loss of training adaptations due to a reduction in or cessation of training, occurs during this period and can affect sports performance in the following season [64,68,69,70].

The kinetics of changes in body composition during the transition period are modulated by its duration, the reduction in the number of training units, and the athletes’ fitness level, potentially leading to unwanted changes [66]. Factors contributing to these changes include fewer training units and less club match play, inactive leisure activities, and the professional football players’ incapacity to adjust their diet to reduced physical activity [71]. An unbalanced diet during this period can lead to an increase in body mass, impacting fitness and performance. Moreover, decreased muscle mass due to a lack of training stimulus during this phase can result in reduced strength and endurance, increasing the risk of injury when intense training resumes during the preparatory period [64].

According to Staśkiewicz et al.’s study, there was an increase in body mass, when the average weight went from 79.27 ± 7.44 kg to 80.13 ± 7.47 kg. An analysis of body composition showed an increase in FM [56]. Reinke et al., in a study involving players of the German top division, showed that, during the transition period, the average body mass of players decreased from 90.2 kg to 88.4 kg; in addition, there was a decrease in LBM on average from 74.5 kg to 72.3 kg and an increase in FM on average from 10.5 kg to 11.2 kg [72]. Ostojic’s study analyzed the body composition of 30 English League 1 players and showed an increase in transition body mass from 74.8 ± 6.0 kg to 77.2 ± 7.6 kg on average. In addition, there was a decrease in LBM from 67.6 ± 5.3 kg to 67.4 ± 6.2 kg and an increase in FM from 9.6 ± 2.5% to 12.6 ± 3.3% [66].

## 6. Nutritional Recommendations and Periodization of Nutrition in Football

To release an expert statement on a variety of nutrition-related topics about professional athletes, the Union of European Football Associations (UEFA) convened specialists in sports nutrition research and practitioners who worked with elite football clubs, national associations, and federations in 2020 [27]. The original statement was released over 10 years ago, but more and more scientific research in the last few years has made an update regarding the timeliness and accuracy of the guidelines necessary [27].

The physical and technical requirements for football have been trending significantly in recent years [73]. As a result, training plans are adjusted appropriately, leading to heavier workloads to help players adjust to demands [12,74,75]. A higher number of matches is linked to a higher risk of injury. In addition, players play matches at varying times to accommodate television schedules. The movement of international players, as well as regional and intercontinental contests, frequently reflect dietary differences in cultures [27].

In nutrition, there is also a great deal of uncertainty. The recommended dietary allowance (RDA), for instance, is the average daily intake that is adequate to meet the nutritional needs of 98% of healthy individuals in a population. However, it is unclear how these values should be interpreted when determining the daily intake of various athlete groups [76].

The “food first” theory—which holds that food is the foundation for the body’s proper operation in an exercise-loaded situation and that supplements can only assist an appropriate nutritional strategy—is endorsed and supported by UEFA-appointed experts [27].

The process of establishing dietary guidelines is complicated by several restrictions. The first has to do with the paucity of studies on football as a sport and the inability of lab models created to mimic the game to accurately recreate match requirements [75,77]. As a result, data from other sports and easier activities must be extrapolated. When it comes to sports like football, participants in recreational football leagues frequently comprise the study group in research studies [12,74]. Results involving elite athletes are rare. Furthermore, the approach taken to evaluate the eating patterns and nutritional status of players is frequently faulty and produces inaccurate data [27].

### 6.1. Energy Processes Occurring during the Game of Football

Football players engage in a range of movements during a game, including walking, sprinting, changing directions, jumping, striking the ball, and making contact with the opposition [78]. Throughout the game, field players’ heart rates are kept at an average of 85% of their maximum heart rates, with an average relative exercise intensity of 70% of their maximum oxygen uptake (VO2 max). Approximately 1300–1600 kcal of energy expenditure is indicated by these values [79,80]. A daily energy expenditure of approximately 3500 kcal was estimated [81,82]. Physical activity is fueled by an integrated series of energy systems that include anaerobic (phosphagen and glycolytic) and aerobic (fat and carbohydrate oxidation) pathways, using endogenous and exogenous substrates [81].

The phosphagen system’s adenosine triphosphate and phosphocreatine offer a quick source of energy for muscle contraction, but not at a level that can sustain an energy supply for longer than about ten seconds [82]. The primary mechanism enabling intense exercise lasting 10–180 s is the anaerobic glycolytic pathway, which quickly metabolizes glucose and muscle glycogen via the glycolytic cascade [79,83].

Aerobic pathways serve as the main source of fuel for events lasting longer than approximately two minutes, as the phosphagen and glycolytic pathways are unable to supply the energy needed for muscles to contract at very high rates. Major substrates include muscle and liver glycogen, intramuscular lipids, adipose tissue triglycerides, and amino acids from muscle, blood, liver, and intestine [27,45].

The body uses more aerobic pathways and fewer anaerobic pathways when oxygen is more readily available to working muscles [27,45]. The body never uses just one pathway, nor does it become more dependent on aerobic pathways overnight. The number of pathways depends on several factors, including substrate availability, previous nutrient intake, training levels, and the intensity, duration, frequency, and type of training [80,83].

Skeletal muscle in athletes is remarkably malleable; this property enables it to react quickly to changes in nutrition and mechanical loads, leading to functional and metabolic adaptations to particular environments [79]. Nutritional recommendations for performance account for these adaptations, stating that energy systems should be modified during different training modalities to guarantee the most efficient use of fuel during exercise [80]. Increases in the number of transport molecules that transport nutrients across membranes to the site of use in the muscle cell, the number of enzymes that activate or regulate metabolic pathways, the strength of the tolerance to metabolic byproducts, and the amount of fuel stored in muscle are examples of adaptations that lead to an increase in metabolic flexibility. Adipose tissue is one substrate that is present in relatively large quantities, while other substrates, like carbohydrate supplementation to replenish glycogen deficiencies after training, may need to be adjusted as needed [27,80].

Muscles use up energy at a rate that is directly correlated with exercise intensity. Working muscles transform chemical energy into kinetic energy and heat during activity, which needs to be expelled by thermoregulatory systems [84]. Muscle cells produce adenosine triphosphate (ATP), the direct source of muscle energy, through the oxidation of carbohydrates and fatty acids. This process produces energy aerobically. Phosphocreatine, the direct source of ATP re-synthesis, decreases significantly from ≈79.0 to ≈20.0 mmol/kg^−1^ dry weight, but the ATP concentration only decreases from ≈25.0 to ≈16.0 mmol/kg^−1^ dry weight, even after intense exercise [85]. It seems that training does not affect the ATP concentration. The phosphocreatine and glycolytic pathways in anaerobic metabolism can also produce ATP [79,84].

### 6.2. Dietary Recommendations for Macronutrient Intake

To achieve sporting goals, nutrition is crucial. Athletes frequently use a range of ergogenic aids in an attempt to increase performance. Any training method, mechanical apparatus, dietary supplement, exercise regimen, pharmacological treatment, or psychological strategy that can boost training adaptation or physical performance is referred to as an ergogenic aid [86]. Ergogenic agents can aid in improving an athlete’s performance during exercise, hastening their recuperation after an activity, and reducing the risk of injury during rigorous training. But often, it would be enough to modify the typical diet to suit the athlete’s needs [78].

Macronutrients do have ergogenic aid effects, even though they cannot be directly categorized as such [87]. An athlete’s performance is improved by consuming carbohydrates right before or right after exercise, because they increase glycogen stores and postpone fatigue [79]. Proteins can optimize body composition, which has an anabolic effect. Furthermore, appropriate growth and development, preserving health and well-being, and lowering the risk of illness, injury, and disease are all impacted by nutrition [88]. A healthy diet helps prevent potential health issues. Poor diet can have an adverse effect on body function, in addition to lowering performance [86,89].

#### 6.2.1. Carbohydrates

The typical pattern of play in team sports involves a ‘stop and go’ style, where players engage in repeated bouts of brief high-intensity exercise interspersed with lower-intensity activity. Carbohydrates serve as a primary energy source for athletic performance in team sports players [90]. As exercise intensity increases, the importance of carbohydrates also rises. However, due to limited carbohydrate reserves in the body, the depletion of liver and intramuscular glycogen during prolonged periods of high-intensity exercise significantly impacts athletic performance. Carbohydrate depletion resulting from prolonged high-intensity team sports impairs performance by inhibiting fat metabolism and leading to the accumulation of metabolites such as ammonia (NH3), lactate, hydrogen ions (H+), and inorganic phosphates (Pi) [91,92] The body’s capacity to store carbohydrates is comparatively small, but it can be significantly altered every day by eating foods that are suggested to help with this [10]. Because they are used in both anaerobic and aerobic pathways, carbohydrates serve as both a flexible substrate for muscle work and a vital source of energy for the brain and central nervous system [93]. They can also support exercise at a variety of intensities. Research indicates that strategies that maintain high carbohydrate availability can help one perform longer, continuous, or intermittent high-intensity exercise more efficiently [94]. A depletion of stores is also linked to fatigue, which manifests as a decreased ability to exercise and a decline in focus. These results serve as the foundation for some dietary approaches that supply carbohydrates before, during, and following physical activity [95].

Carbohydrate loading, whether achieved through a high-carbohydrate diet or additional carbohydrate supplements following the depletion of carbohydrate stores due to prolonged high-intensity exercise, enhances glycogen synthesis by increasing concentrations of glucose transporter (GLUT-4) and glycogen synthase enzymes [96]. Furthermore, it elevates the secretion of anabolic hormones such as growth hormone (GH) and testosterone and enhances glucose efficiency through heightened insulin secretion as the blood glucose concentration increases [90]. Additionally, it lowers levels of free fatty acids (FFA) and glycerol in the bloodstream. Moreover, the consumption of additional carbohydrate supplements before exercise enhances carbohydrate metabolic efficiency by activating the intramuscular pyruvate dehydrogenase complex (PDC) [97], leading to an increase in intramuscular adenosine triphosphate (ATP), creatine phosphate (PCr), and glycogen content through heightened excess post-exercise oxygen consumption (EPOC) [98]. These benefits of carbohydrate loading via a high-carbohydrate diet or additional carbohydrate supplements have been demonstrated to optimize liver and muscle glycogen storage and improve exercise performance across various athletes [98].

Athletes are advised to up their carbohydrate intake to a value of 6–8 g/kg body weight the day before, the day of, and the day after the competition, due to the significance of muscle glycogen in post-match preparation and recovery. It is important to remember that, even 48 h after the match, the type II fibers’ muscle glycogen content might not fully recover with an intake of about 8 g per kilogram of body weight [99]. Alternatively, a daily carbohydrate intake of 3 to 6 g/kg body weight may be sufficient for energy replenishment and recovery, given the lower daily loads on typical training days (one session per day in a microcycle with one match day), combined with the fact that athletes typically do not perform any additional physical activity outside of the club [46]. On training days, the daily carbohydrate intake should range from 3 to 8 g/kg body weight per day, depending on the training scenario, match schedule, and player-specific training goals [27,83].

#### 6.2.2. Proteins

The musculoskeletal and tendon tissues are stressed by systematic football training; therefore, protein-containing structure repair is necessary to preserve and enhance their integrity and function [94]. Skeletal muscle damage caused by physical exertion (EIMD) is associated with increased proteolysis and protein breakdown [100], tissue damage at the membrane and subcellular levels (both within and outside the sarcomeres), and the release of pro-inflammatory cytokines (such as IL-1 and IL-6) by muscles and other tissues to mobilize local and systemic immune reserves [101]. Some nutritional interventions suggest that an increased intake of protein and carbohydrates may mediate regeneration processes and promote muscle recovery, thus expediting the return to full functionality [46]. Protein supplementation has been shown to accelerate skeletal muscle protein turnover by disrupting their synthesis and degradation under conditions of increased physiological stress, such as those experienced during intense football matches [102].

An RDA level of 0.8 g/kg body weight per day is required for the majority of Europeans [94]. According to the available research, 1.6–2.2 g of protein per kilogram of body weight should be consumed daily to enhance training adaptation [103]. A higher intake might only make sense during brief bursts of intense training or when consuming less energy [104]. When there is an adequate energy supply to meet training needs, a mixed diet will enable protein requirements to be satisfied. Athletes typically report consuming the recommended amount of protein [46]. A balanced diet that includes moderate servings of high-quality protein spread throughout the day should be the goal for achieving the recommended daily intake of protein [27]. Most football players can avoid using protein supplements if they plan their diets properly, but they are a quick and easy way to replace food, particularly after a workout [27]. An athlete should aim to consume ~0.4 g/kg body weight of protein per meal, or 3–4 meals a day, to reach a daily total of ~1.6 g/kg body weight of protein [103]. Leucine content is one of the factors that affect protein quality, which is a significant concern. Leucine is an amino acid that stimulates the synthesis of muscle fibers; approximately 2.5 g of leucine should be consumed with each meal [105]. Because of the catabolic environment that the energy deficit creates, protein requirements rise during energy restriction above the recommended RDA values. Thus, depending on the training load and other metabolic stresses, such as weight loss or injury recovery, it makes sense to increase protein intake in this case to a value of 2.0–2.4 g/kg body weight per day [106].

#### 6.2.3. Fats

A balanced diet must include fat because it gives you energy, forms part of cell membranes, and makes it easier for you to absorb fat-soluble vitamins [107]. For individuals who are not physically active, 10% of their total dietary energy intake should come from an adequate intake of linoleic and α-linolenic acid [108].

Eicosapentaenoic acid (EPA) and docosahexaenoic acid (DHA) are omega-3 polyunsaturated fatty acids that play a crucial role in regulating inflammatory processes in the body [109]. These fatty acids have been suggested to offer benefits to athletes due to their anti-inflammatory properties. These benefits include replacing arachidonic acid in membrane phospholipids, influencing the release of inflammatory eicosanoids and cytokines, and producing anti-inflammatory lipid mediators. Emerging evidence among athletes indicates that improving omega-3 status through supplementation can promote muscle recovery, reduce oxidative stress associated with exercise, and lower levels of neurofilament light, a biomarker for axonal injury, following sport-related subconcussive head injuries [110,111].

To meet overall energy needs, athletes should modify their fat intake to accommodate protein and carbohydrate requirements. Guidelines for reducing the consumption of trans fats and using saturated fats sensibly should also be adhered to. As a result, the daily calorie value of the food is often consumed as fat, ranging from 20 to 35 percent [107]. Athletes should customize their fat intake according to their exercise load and desired body composition [27]. Some athletes purposefully consume a smaller amount of fat-containing products. This causes the consumption of fat to be restricted to less than 15–20% of daily energy and causes many products that contain important nutrients to be avoided [112]. The popularity of the ketogenic diet is another significant factor. There are currently no intervention or observational studies on team sports and their applicability [113]. A diet high in fat and low in carbohydrates is not advised for football players due to a lack of evidence [112].

#### 6.2.4. Fluids

Athletes sweat 1.5 L per hour, consume 0.7 L per hour, and change in body mass by 1.5% on average [114].

Sweat evaporating through the skin’s surface is the main way that the body loses heat during increased physical activity [115]. This can result in sweat-induced dehydration and is a vital mechanism for regulating internal body temperature. A 2% body weight water deficit is referred to as 2% dehydration. While there are other factors, such as the loss of water vapor and carbon dioxide with exhaled air, the primary cause of body mass reduction during intense physical activity is sweating due to thermoregulation [114,116].

Even mild dehydration of 1% to 2% can impair football-specific performance, particularly intermittent high-intensity sprinting and dribbling skills. Common symptoms of dehydration among football players include a dry, sticky mouth, fatigue, thirst, and reduced urine production. In addition to water loss, sweating also leads to the depletion of electrolytes, notably sodium and chloride [116].

Sweating also causes the loss of electrolytes, mainly sodium. During football matches, players can lose a significant amount of sodium, ranging from 30 mmol/L to 62 mmol/L, which translates to approximately 3.9 to 6.1 g of salt. Sweating can result in a fluid loss of approximately 30 g per minute, which can exceed 1.8 kg per hour. Along with water, important electrolytes for muscle contraction, such as sodium (Na), chloride (Cl), and potassium (K), are also lost. The loss of sodium and chloride tends to be greater than potassium according to various studies [117,118]. Hydration and electrolyte replenishment are related, because hydration increases the amount of fluid that is retained after ingestion [119].

Dehydration can be detected through the evaluation of urine-specific gravity. Values greater than 1020 g/mL indicate dehydration [116]. To estimate the amount of fluid lost during exercise, it is also critical to weigh oneself both before and after. Preventing a loss of more than 2% of body mass before, during, and after exercise is the aim of rehydration. This will reduce the likelihood of exercise capacity being lost [115].

Hydration is crucial to the recovery process after exercise. Athletes should try to replace all lost fluid and electrolytes before the next training session or game if they notice a decrease in body mass. For every kilogram of body mass lost during exercise, the body should rehydrate with approximately 1.5 L of fluids [115,119].

#### 6.2.5. Pre-, During-, and Post-Exercise Nutrition

Pre-game nutrition is essential for football players to stabilize blood sugar levels, preserve glycogen stores, and maintain hydration. Inadequate intake can lead to early glycogen depletion, hypoglycemia, fatigue, and reduced performance [120]. Carbohydrate intake, ideally 7–10 g/kg of body mass per day, plays a vital role in glycogen synthesis. It is crucial to consume carbohydrate-rich meals providing at least 1.0 g/kg of body mass 3–4 h before the match [121]. While protein consumption is not critical pre-match, meals should include low-GI, complex carbohydrate-rich foods to stabilize blood glucose levels. Unsuitable meals high in fiber, protein, and fat should be avoided, as they take longer to digest and are not suitable before high-intensity activity [120].

During matches, the dietary focus is on sustaining blood glucose and muscle glycogen to prolong energy production and delay fatigue. An optimal fluid intake, easily absorbed without discomfort, is crucial. Halftime provides an opportunity to replenish fluids and carbohydrates lost during the game. Isotonic sports drinks or carbohydrate supplementation (30–60 g/h) facilitate quick digestion, hydration, and fueling to maintain performance [122].

Recovery nutrition aims to replenish glycogen stores and aid muscle repair. Whey protein, due to its rapid absorption, is preferred post-exercise. Ingesting carbohydrates (1.0–1.2 g/kg body mass per hour) as small snacks every 15–30 min for up to 4 h post-match enhances glycogen replenishment, insulin secretion, and muscle recovery [123]. Combining carbohydrates and protein enhances performance, with sprint and time trial performance improving after intense exercise. Post-exercise meals should include high-glycemic-index carbohydrate sources and an optimal protein intake to promote muscle protein synthesis. A post-match meal within four hours should maintain a 3:1 carbohydrate-to-protein ratio to prevent reduced calcium release from the muscle sarcoplasmic reticulum in glycogen-depleted conditions [124].

## 7. Supplementation in Football

As per the International Olympic Committee’s definition, a dietary supplement encompasses any food, food component, nutrient, or non-food substance deliberately consumed alongside the regular diet to achieve specific health or performance advantages [125]. Athletes engage in supplement use for diverse reasons, often influenced by manufacturer advertising [126]. The supplement industry is primarily profit-driven and caters to consumer demand and preferences. Frequently, there is insufficient scientific proof supporting the efficacy of individual substances and the tangible benefits derived from supplementation [127].

To streamline information verification for athletes, coaches, and sports organizations, the Australian Institute of Sport (AIS) has devised a supplement classification system with four categories: A, B, C, and D [128]. This system categorizes supplements based on scientific evidence and practical considerations to ascertain their safety, permissibility, and efficacy in enhancing sports performance. Category A includes supplements with validated effectiveness and safety, category B comprises supplements requiring further research to establish their efficacy, category C encompasses supplements lacking evidence of efficacy, and the final category, D, comprises substances considered unsafe due to anti-doping concerns [128].

Footballers’ dietary plans should prioritize natural food sources over supplements, reserving the latter for specific health or performance targets. The duration and nature of supplementation should be overseen by a qualified dietitian or sports physician [125]. The demands of regular training and matches can escalate the need for vitamins, macronutrients, and micronutrients among professional footballers, crucial for supporting their body’s metabolic functions [27]. Ensuring a sufficient intake of B vitamins and antioxidants like vitamins A, C, and beta-carotene, along with vitamin D, iron, calcium, and magnesium, is paramount [127,129,130].

Under unique circumstances, such as adhering to a low-energy diet for weight loss, eliminating certain food groups, or having irregular eating patterns, there may be a risk of inadequate vitamin and mineral intake. In such cases, supervised supplementation under the guidance of a dietitian may be warranted to meet 100% of the recommended daily allowances (RDAs) for all essential nutrients. Nevertheless, the primary focus should remain on obtaining the necessary nutrients from whole foods before considering supplementation [131].

The UEFA experts, in their statement, identify several substances that could offer potential advantages for soccer players, including caffeine, creatine, ß-alanine, and nitrates, all categorized as Group A by the AIS [27,128].

The performance-enhancing properties of caffeine are widely recognized, with research confirming its positive effects on performance during intermittent, endurance, and resistance exercises [132]. There is evidence suggesting that caffeine consumption can have beneficial effects on outcomes in team sports, with data indicating improvements in the crucial physical and technical performance aspects required for football. Ingesting caffeine at a dosage of 2–6 mg/kg of body weight has been shown to enhance performance in repeated sprints and jumps, reactive agility, jump height, and passing accuracy [73]. Presently, caffeine is under scrutiny by the World Anti-Doping Agency (WADA), and there is the potential for caffeine to be reclassified as a banned substance [133].

Creatine supplementation stands out as one of the most extensively studied and beneficial ergogenic aids for athletes. Increasing muscle creatine stores can enhance exercise performance and facilitate training adaptation [134]. Research indicates that supplementing with creatine can lead to improved performance in high-intensity exercises, thereby enhancing overall training outcomes [135]. Moreover, creatine supplementation accelerates glycogen replenishment rates, which can be advantageous for athletes undergoing prolonged submaximal exercise or engaging in repetitive high-intensity workouts, especially concerning aerobic and anaerobic metabolism [136]. A typical outcome of creatine supplementation is a weight gain of around 1–2 kg, primarily attributed to water retention, with no adverse health effects noted when following appropriate supplementation protocols [137].

ß-alanine enjoys widespread popularity as an ergogenic aid, with approximately 61% of team sport athletes incorporating it into their regimen [138,139]. Supplementation with ß-alanine can elevate muscle carnosine levels, bolster intramuscular buffering capacity, and enhance performance in intermittent high-intensity exercises, all while complying with the regulations set forth by the World Anti-Doping Agency [139]. Its supplementation represents a viable nutritional strategy to combat fatigue and optimize performance, especially in scenarios involving intense efforts where glycolytic pathways are prominently engaged, leading to significant hydrogen ion accumulation [138]. Although temporary paraesthesia may occur as a side effect of ß-alanine supplementation, it does not pose any adverse health risks [140].

Nitrates and nitrites have conventionally been viewed as byproducts of nitric oxide (NO) metabolism. However, recent findings propose that nitrates might act as precursors to nitric oxide, transitioning from nitrate to nitrite and eventually to nitric oxide [141]. Nitric oxide is synthesized in the body through the oxidation of l-arginine, facilitated by nitric oxide synthase (NOS). It plays a vital role in regulating skeletal muscle function, enhancing exercise performance by reducing ATP expenditure for muscle contraction, improving the efficiency of mitochondrial respiration, and boosting the blood flow to muscles [142]. Individual optimal dosages vary, but benefits typically manifest within 2–3 h after administering a nitrate bolus containing 5–9 mmol (310–560 mg). Prolonged periods of nitrate intake (>3 days) may also enhance performance, particularly among well-trained athletes [141,143].

## 8. Conclusions

The present study examines the concept of nutrition periodization in the context of football training, demonstrating the validity of using an individualized approach to managing players’ diets. Crucial to optimizing the athletic performance and body composition of players is adjusting the amount of energy provided, macronutrient ratios, and hydration depending on the phase of the training macrocycle and individual athletic and health goals.

During the preparatory period, where the focus is on the development of strength and muscle mass, increased energy and protein requirements are critical to support anabolic processes and muscle adaptations. During the competitive period, dominated by match loads and higher-intensity training, the need for a higher carbohydrate intake to replenish muscle glycogen and support rapid recovery is emphasized. During the transition period, which often serves a regenerative function, an adequate adjustment of energy and macronutrient intake can counteract undesirable changes in body composition, such as fat gain. In the context of fluids, adequate hydration is important during any training phase, but especially during the competitive phase, where fluid losses are intensified by frequent matches and training in high temperatures.

An analysis of the impact of nutritional periodization on the body composition of football players indicates its fundamental importance in modulating SMM, FM, and overall physical condition. Adapting nutritional strategies, such as meal timing, energy balance, and macronutrient selection, allows precise control over changes in body composition, which directly translates into athletic performance and the ability to quickly regain game readiness.

An integral part of sports preparation should be close cooperation between coaches, nutritionists, and athletes to continuously adapt and optimize nutrition, which is as important as well-planned training.

## Figures and Tables

**Figure 1 nutrients-16-01332-f001:**
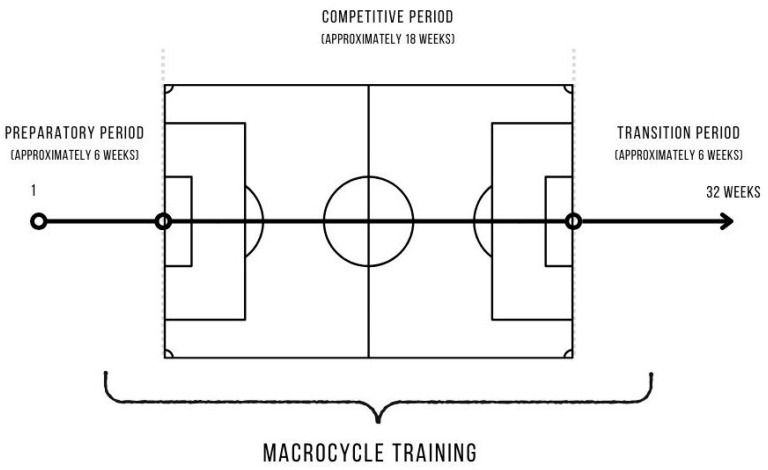
Macrocycle training in football (taking into account the two-cycle model) [41].

## Data Availability

Not applicable.

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
