# Peer review of "Changes in Body Composition and Nutritional Periodization during the Training Macrocycle in Football—A Narrative Review"

_nutrients, 2024, doi:10.3390/nu16091332_

Round 1
Reviewer 1 Report
Comments and Suggestions for Authors
Some recommendations and questions are provided:
Abstract – should include what data are collected, how was data collected, analysis done. Results derived from the analyses
Line 41 to 43 – 2 sentences typically does not constitute a paragraph (same with lines 154 to 156) – and subsequent sections in the paper.
After the introduction – provide the research objective clearly
Was not able to find the method section
If this paper is a documentary analysis or literary review, there should be some scope – limitations, inclusion criteria, exclusion criteria
Importantly, there should be a sort of guiding framework for the discussion and not just providing a series of literature
Author Response
Comment 1.
Abstract – should include what data are collected, how was data collected, analysis done. Results derived from the analyses.
Thank you very much for your valuable comment. We have corrected the introduction. We hope it is appropriate for you.
Comment 2.
Line 41 to 43 – 2 sentences typically does not constitute a paragraph (same with lines 154 to 156) – and subsequent sections in the paper.
Corrected as suggested by the reviewer.
Comment 3.
After the introduction – provide the research objective clearly
Thank you very much for your valuable comment. We have added the section you asked for.
Comment 4.
Was not able to find the method section
Thank you very much for your comment, we have added section 2. We hope it is sufficient for you.
Comment 5.
If this paper is a documentary analysis or literary review, there should be some scope – limitations, inclusion criteria, exclusion criteria
Missing information in the manuscript was completed.
Comment 6.
Importantly, there should be a sort of guiding framework for the discussion and not just providing a series of literature
Corrected in text.
Thank you so much for taking the time to evaluate our work. We have tried to incorporate all your valuable suggestions. If we could improve our work in any way, please let us know.
Kind regards,
Authors.
Reviewer 2 Report
Comments and Suggestions for Authors
The present manuscript details important information that is highly relevant to a large number of both competitive and recreational football players. While the topics discussed are logical, their narrative framework is somewhat piecemeal and otherwise redundant. Moreover, many sections lack depth and thus extensive editing should be performed before this manuscript can appropriately demonstrate its value in the present journal.
General Comments:
Throughout this manuscript, passive rather than active voice is implemented, which makes the narrative somewhat cumbersome to read on a sentence-by-sentence level. I would suggest editing it to improve readability in this way and to add someone with particular English language-focus if possible to assist with other related suggestions.
One of my main concerns is that I see several sections throughout that are redundant information. For example, Page 6, Lines 230-243 contain very little new information relative to section 4 and this trend occurs throughout. Thus, a large majority of this review should be truncated to minimize repeated information.
I think it may be beneficial with respect to reducing redundancy to combine the sections for the nutritional and body composition sections for each phase. This would be sensible since body composition not only affects performance but also body composition. Therefore, you would have both considerations tied together for transition, competitive, and preparatory phases.
It is unfortunate that supplements are a consideration for football athletes mentioned early in the manuscript but no details to commonly used supplements outside of electrolyte- or macronutrient- containing beverages are mentioned. It would be relevant to consider how common sports supplements such as bicarbonate, caffeine, and/or creatine supplementation could be relevant given their formative role in athletic endeavors.
The sections on macronutrient considerations lack considerable detail. I wish there was more on exercise physiology-related background with respect to how blood glucose and muscle glycogen use increases as a function of increased intensity/ relative VO2max percentage. Furthermore, I do not see anything on timing importance, especially with respect to protein and carbohydrate pre- and post-training. I’m also curious why the essential value of polyunsaturated fatty acids like N-6 and especially N-3 fatty acids are not mentioned when many other broad nutritional recommendations are discussed. I also find it odd that although there may be a paucity of data on football-specific athletes, there is no previous data mentioned about other recreational athletes with similar bioenergetic demands and their nutritional considerations. Overall, this section reads as entirely too generalized to the non-athletic public.
Specific Comments
Page 2, line 50: It would be appropriate to include a purpose statement that ultimately drove the impetus of this manuscript.
Page 2, Line 53: Some cases such as powerlifting and weightlifting make the term “always” a bit bold.
Page 2, Lines 56-57: This sentence is redundant to the prior.
Page 2, Lines 59-61: This sentence is likewise redundant.
Page 3, Lines 100-111: I only see the two-cycle variation mentioned and not the single nor three-cycle versions.
Page 3, Line 123: Please provide examples of these “technical and tactical exercises”
Page 4: Lines 139-142: Is there anything to add on how this miscellaneous volume is managed when the assumption is the athletes are net recuperating?
Page 4, Lines 142-144: This sentence is awkward and should be reworded.
Page 4: Lines 140-141: “…injuries and microinjuries…” might be better rephrased to “injuries of various severity” or similar.
Page 4, Line 148: Does “dissertation” refer to the transition period? This term is a bit odd here.
Page 4, Lines 150-153: Again, this sentence reads as redundant.
Page 4, Lines 153-155: Redundant and unnecessary, especially as a separate paragraph.
Page 4, Lines 157-166: This is all great information but there is no context or explanation as to how this impacts recovery.
Page 4, Line 173: What kinds of exercises are typically prescribed and what relative intensity (%1RM, RPE, etc.)
Page 5, Lines 179-185: Perhaps this entire section should be omitted since it is addressed later in the nutritional consideration
Page 5, Lines 187-193: Perhaps it would be best to address what relevant body composition for football looks like (skeletal muscle increases, fat reductions, etc.) here and not so much later as it is.
Page 5, Line 214: What are “relative” values?
Page 5, Lines 217-219: The information provided here makes me wonder if athletes self-select into their given positions and if there is a need to maintain a more general body composition in the event that a player plays more than one position?
Page 8, Lines 362-406: This information is all great but should be made more succinct and relegated to section 3 to provide context within the physical requirements section. Much of it – such as lines 390-396 – also aren’t overly relevant to the purpose of this review with the level of detail contained.
Page 10, Lines 467-468: Is this referring to the RDA or the 1.6-2.2g/kg?
Page 11, Line 499-500: More detail would be nice about this more than just general “electrolytes”? Why is no information addressed on specific sodium, potassium, chloride, etc. recommendations?
Comments on the Quality of English LanguageAlthough organizationally redundant and often cumbersome, language is a minor issue
Author Response
Thank you so much for taking the time to evaluate our work. We have tried to incorporate all your valuable suggestions. If we could improve our work in any way, please let us know.
Comment 1.
Throughout this manuscript, passive rather than active voice is implemented, which makes the narrative somewhat cumbersome to read on a sentence-by-sentence level. I would suggest editing it to improve readability in this way and to add someone with particular English language-focus if possible to assist with other related suggestions.
Thank you very much for your comment. We hope that the manuscript in its current form meets your expectations.
Comment 2.
One of my main concerns is that I see several sections throughout that are redundant information. For example, Page 6, Lines 230-243 contain very little new information relative to section 4 and this trend occurs throughout. Thus, a large majority of this review should be truncated to minimize repeated information.
Thank you very much for your suggestion. We have made every effort to correct the shortcomings in the manuscript according to your comments.
Comment 3.
I think it may be beneficial with respect to reducing redundancy to combine the sections for the nutritional and body composition sections for each phase. This would be sensible since body composition not only affects performance but also body composition. Therefore, you would have both considerations tied together for transition, competitive, and preparatory phases.
Thank you very much for your suggestion. We have changed the elements of the manuscript according to your advice. We believe that this has had a positive effect on the clarity of the text.
Comment 4.
It is unfortunate that supplements are a consideration for football athletes mentioned early in the manuscript but no details to commonly used supplements outside of electrolyte- or macronutrient- containing beverages are mentioned. It would be relevant to consider how common sports supplements such as bicarbonate, caffeine, and/or creatine supplementation could be relevant given their formative role in athletic endeavors.
We have added a separate chapter on supplementation. We hope it is sufficient for you.
Comment 5.
The sections on macronutrient considerations lack considerable detail. I wish there was more on exercise physiology-related background with respect to how blood glucose and muscle glycogen use increases as a function of increased intensity/ relative VO2max percentage. Furthermore, I do not see anything on timing importance, especially with respect to protein and carbohydrate pre- and post-training. I’m also curious why the essential value of polyunsaturated fatty acids like N-6 and especially N-3 fatty acids are not mentioned when many other broad nutritional recommendations are discussed. I also find it odd that although there may be a paucity of data on football-specific athletes, there is no previous data mentioned about other recreational athletes with similar bioenergetic demands and their nutritional considerations. Overall, this section reads as entirely too generalized to the non-athletic public.
Supplemented in the text of the manuscript.
Page 2, line 50: It would be appropriate to include a purpose statement that ultimately drove the impetus of this manuscript.
Added as suggested.
Page 2, Line 53: Some cases such as powerlifting and weightlifting make the term “always” a bit bold.
Changed according to the comment to "usually"
Page 2, Lines 56-57: This sentence is redundant to the prior.
Changed as suggested by the reviewer.
Page 2, Lines 59-61: This sentence is likewise redundant.
Changed as suggested by the reviewer.
Page 3, Lines 100-111: I only see the two-cycle variation mentioned and not the single nor three-cycle versions.
Clarified according to the reviewer's comment.
Page 3, Line 123: Please provide examples of these “technical and tactical exercises”
Added as suggested.
Page 4: Lines 139-142: Is there anything to add on how this miscellaneous volume is managed when the assumption is the athletes are net recuperating?
I don't understand your suggestion. Please explain to me what it is about and how I can make the changes.
Page 4, Lines 142-144: This sentence is awkward and should be reworded.
Changed as suggested by the reviewer.
Page 4: Lines 140-141: “…injuries and microinjuries…” might be better rephrased to “injuries of various severity” or similar.
Changed as suggested by the reviewer.
Page 4, Line 148: Does “dissertation” refer to the transition period? This term is a bit odd here.
Changed to "detreining." Thank you for your suggestion.
Page 4, Lines 150-153: Again, this sentence reads as redundant.
Changed as suggested by the reviewer.
Page 4, Lines 153-155: Redundant and unnecessary, especially as a separate paragraph.
Changed as suggested by the reviewer.
Page 4, Lines 157-166: This is all great information but there is no context or explanation as to how this impacts recovery.
Changed in text. As suggested, the impact of the changes described in the paragraph has been explained.
Page 4, Line 173: What kinds of exercises are typically prescribed and what relative intensity (%1RM, RPE, etc.)
Changed as suggested by the reviewer.
Page 5, Lines 179-185: Perhaps this entire section should be omitted since it is addressed later in the nutritional consideration
Changed as suggested by the reviewer.
Page 5, Lines 187-193: Perhaps it would be best to address what relevant body composition for football looks like (skeletal muscle increases, fat reductions, etc.) here and not so much later as it is.
Changed as suggested by the reviewer.
Page 5, Line 214: What are “relative” values?
Relative values, e.g. %FM instead of FM expressed in kilograms. The absolute values of taller players will be higher than those of shorter players, so relative values should be compared. Clarified in the text.
Page 5, Lines 217-219: The information provided here makes me wonder if athletes self-select into their given positions and if there is a need to maintain a more general body composition in the event that a player plays more than one position?
Definitely, it's a complex issue. In the case of athletes, the choice of position can result from both individual physical predispositions and the team's needs. During the growing process, physical characteristics can be crucial in determining which position a player is best suited for. However, even if an athlete has predispositions to play in a specific position, the position itself may require a specific body composition that supports the specific skills associated with that position. For example, a player playing as a central striker in football may need greater strength and jumping ability, while a player playing on the wing may require speed and agility. Therefore, to be versatile and effective players, some athletes may strive to maintain a more general body composition, allowing them flexibility in playing different positions. However, the final choice of position and body composition depends on many factors, including individual predispositions, team strategy, and the requirements of the specific sport.
Page 8, Lines 362-406: This information is all great but should be made more succinct and relegated to section 3 to provide context within the physical requirements section. Much of it – such as lines 390-396 – also aren’t overly relevant to the purpose of this review with the level of detail contained.
Corrected according to the comment.
Page 10, Lines 467-468: Is this referring to the RDA or the 1.6-2.2g/kg?
It increases above the recommended values at the RDA level. Corrected in text.
Page 11, Line 499-500: More detail would be nice about this more than just general “electrolytes”? Why is no information addressed on specific sodium, potassium, chloride, etc. recommendations?
Added as suggested.
Thank you very much for your insightful review. We have incorporated all your comments in the revised manuscript. We believe that your advice significantly influenced the quality of our manuscript. Please let me know if we can improve it in any way.
Kind regards,
Authors.
Round 2
Reviewer 1 Report
Comments and Suggestions for Authors
Evidence that the author/s already made substantial improvement of the paper.
However, just a minor issue of single sentence paragraph, lines 70 - 72
lines 92 - 93 - please check entire text